# A Convergence Analysis of Gradient Descent on Graph Neural Networks

**Pranjal Awasthi**
Google Research
pranjalawasthi@google.com

**Abhimanyu Das**
Google Research
abhidas@google.com

**Sreenivas Gollapudi**
Google Research
sgollapu@google.com

## Abstract

Graph Neural Networks (GNNs) are a powerful class of architectures for solving learning problems on graphs. While many variants of GNNs have been proposed in the literature and have achieved strong empirical performance, their theoretical properties are less well understood. In this work we study the convergence properties of the gradient descent algorithm when used to train GNNs. In particular, we consider the realizable setting where the data is generated from a network with unknown weights and our goal is to study conditions under which gradient descent on a GNN architecture can recover near optimal solutions. While such analysis has been performed in recent years for other architectures such as fully connected feed-forward networks, the message passing nature of the updates in a GNN poses a new challenge in understanding the nature of the gradient descent updates. We take a step towards overcoming this by proving that for the case of deep linear GNNs gradient descent provably recovers solutions up to error $\epsilon$ in $O(\log(1/\epsilon))$ iterations, under natural assumptions on the data distribution. Furthermore, for the case of one-round GNNs with ReLU activations, we show that gradient descent provably recovers solutions up to error $\epsilon$ in $O(\frac{1}{\epsilon^2} \log(\frac{1}{\epsilon}))$ iterations.

## 1 Introduction

In the last decade, deep neural networks have been successfully used for a variety of machine learning tasks. In particular, optimization of various deep neural network architectures using gradient descent has been empirically (and in some cases, theoretically) shown to work surprisingly well, with an ability to obtain small training error on a variety of problems and datasets. In this paper, we focus on a particular class of neural network architectures known as Graph Neural Networks (GNNs).

Graph Neural Networks (GNNs) have become a popular choice for machine learning tasks with graph structured data [Hamilton et al., 2017, Kipf and Welling, 2016, Veličković et al., 2017]. Computation in GNNs is performed by each node sending and receiving messages along the edges of the graph, and aggregating messages from its neighbors to update its embedding vector. After a few rounds of message passing, the computed node embeddings from all the nodes are aggregated to compute the final output [Gilmer et al., 2017]. This leads to a simple and elegant architecture for a variety of graph-related problems in practice.

While theoretically understanding the optimization, learning, and generalization properties of deep neural networks is a challenging task in itself, our current understanding of GNNs lags significantly behind their more popular counterparts such as fully connected feed-forward networks and convolutional neural networks (CNNs). In the context of GNNs recent theoretical works have focused on questions such as as the representation power of various GNN architectures [Xu et al., 2019b,a, Loukas, 2020a,b] and establishing generalization bounds [Garg et al., 2020]. However, the optimization properties of GNN architectures are less explored. These involve identifying conditions under which algorithms like gradient descent can be shown to be provably effective in optimizing GNNs.

35th Conference on Neural Information Processing Systems (NeurIPS 2021).

Such analysis has been carried out in recent years for fully connected feed-forward networks and CNNs, mainly restricted to depth-2 non-linear networks or deep linear networks [Ge et al., 2017, Andoni et al., 2014, Bartlett et al., 2017, Arora et al., 2018a, Hardt and Ma, 2016]. However, the message passing nature of GNNs makes it harder to theoretically analyze the optimization problem in GNNs, compared to other neural network architectures.

We provide the first convergence analysis of gradient descent for GNNs, and provably show that gradient descent can recover near optimal solutions. We first consider the case of a one-round (or one hidden-layer) GNN with ReLU activations. This scenario already captures much of the complexity of the problem, in particular allowing for multiple local optima. Following standard assumptions made in the case of analyzing feed-forward networks, we show that when the input data distribution is the standard Gaussian distribution, and the training objective is to minimize least square regression loss, gradient descent converges to the population error of $\epsilon$ in $O(\frac{1}{\epsilon^2} \log(\frac{1}{\epsilon}))$ iterations. In particular, our analysis applies to the standard practice of initializing the network with weights from a Gaussian distribution, and we do not perform any complicated pre-processing step to initialize the network.

We then extend our result to the multi-round (or multi-layer) case, but for the case of linear activations (analogous to the setting for deep linear networks [Arora et al., 2018a, Bartlett et al., 2018]). In this case, we show a stronger convergence result; gradient descent converges to an error $\epsilon$ in $O(\log(1/\epsilon))$ iterations. All our results are for the realizable setting, i.e, we assume that the data is indeed generated by an (unknown) underlying GNN, as is typical in analysis of gradient descent for other network architectures (see the discussion of related work in Section A). The rest of the paper is organized as follows. In Section 2 we set up notation and discuss the model of GNNs that we consider. In Section 3 we present our first main convergence theorem for the case of one layer/round GNNs with ReLU activations. This is followed by our second main result on analyzing deep linear GNNs in Section 4. We conclude with discussion and open problems in Section 6. We discuss the most relevant related works in individual sections (Section 3 and Section 4), and provide a more comprehensive survey of related work in Section A.

## 2 Preliminaries

Graph Neural Networks operate via passing and aggregating messages among the nodes of a graph. Given an undirected unweighted graph $G = (V, E)$, let $n = |V|$ be the number of nodes in the graph and $m = |E|$ be the number of edges. Initially each node $i \in [n]$ is initialized with its own private embedding $x_i \in \mathbb{R}^r$, where $r$ denotes the embedding size. Then the computation in a GNN proceeds in a number of rounds where at round $\ell$, the new embedding $x_i^\ell$ for vertex $i$ is obtained via a combination of *aggregate* and *combine* steps Gilmer et al. [2017] as shown below.

$$a_i^{(\ell)} = \text{AGGREGATE}(\{x_j^{\ell-1} : j \in N(i)\}) \tag{1}$$

$$x_i^{(\ell)} = \text{COMBINE}(x_i^{(\ell-1)}, a_i^\ell). \tag{2}$$

Here $N(i)$ is the neighborhood of vertex $i$. When comparing to standard architectures such as feed-forward networks, it is useful to think of a round of message passing as computation performed by one layer of a more standard architecture. Hence an $\ell$-round GNN shouldbe thought of as a network with $\ell$ hidden layers. Different choices of the aggregate and combine operations lead to different versions of GNNs such as graph convolutional networks (GCNs) [Kipf and Welling, 2016], GraphSAGE [Hamilton et al., 2017], and graph isomorphism networks (GINs) [Xu et al., 2019b] to name a few. The aggregate operation is typically a simple pooling operation such as the sum or average, and the combine operation is implemented via a low depth neural network. Furthermore, in the most popular implementation, i.e, GCNs, the network parameters are shared across the different rounds. This setting will also be the focus of study in our work. In particular we consider two problem settings. In the first case (Section 3) we assume that the number of rounds in the GNN is one (i.e. one hidden layer) and that the combine operation is a depth one network with ReLU activations. In the second setting (Section 4) we consider arbitrary round $L$ GNNs but restrict the combine operation to be a linear network. In both the settings we will assume that the aggregate operation is a sum, and that the initial input embeddings for the nodes are drawn from the standard Gaussian distributions, i.e., $N(0, I_{r \times r})$. Finally, our analysis for the case of ReLU activations will rely crucially on the notion of dual activations and their properties that we recap below.

**Definition 1** ([Daniely et al., 2016]). *The dual activation of $\sigma$ is the function $\hat{\sigma} : [-1, 1] \mapsto \mathbb{R}$ defined as*

$$\hat{\sigma}(\rho) = \mathbb{E}[\sigma(X)\sigma(Y)], \tag{3}$$

*where $X$ and $Y$ are jointly Gaussian random variables with mean, zero variance one, and covariance $\rho$.*

The work of Daniely et al. [2016] showed that dual activations satisfy many nice properties such as continuity in $[-1, 1]$ and the fact that they are convex in $[0, 1]$. For a more extensive list of the properties of the dual activations please refer to Lemma 11 in Daniely et al. [2016].

## 3 One round GNNs with ReLU activations

In ths section we present our main result on convergence of gradient descent for learning an unknown one round GNN with ReLU activations. We consider a graph $G = (V, E)$ with $n$ nodes and maximum degree $d$. Furthermore, we assume that there is an unknown GNN generating outcomes as

$$y = \sum_{i=1}^{n} \sigma(W^* \bar{x}_i), \tag{4}$$

where $\bar{x}_i = \sum_{j \in N(i)} x_j$ and each $x_j$ is drawn i.i.d. from $N(0, I_{r \times r})$, and $\sigma(t) = \sqrt{2} \max(t, 0)$. In other words, the aggregate operation is a sum, and the combine operation is a depth-1 neural network with ReLU activations that produces an $h$-dimensional embedding for each node, where $h$ is the number of hidden units (inner dimensionality of $W^*$). Finally, the embeddings of all the nodes are summed up to produce an $h$-dimensional output for the graph $G$.

The $\sqrt{2}$ factor multiplication to the ReLU function is for technical convenience and does not affect our results. The ground truth matrix $W^*$ is an $h \times r$ matrix where we denote $w_1^*, \dots, w_h^*$ to be the rows of $W^*$. Without loss of generality we will assume that $\|w_j^*\| = 1$, and again our results extend easily to the case when $\|w_j\|$ is bounded for all $j$ (see Appendix B). We train another one round GNN with unknown parameter matrix $W$ to minimize the following loss via gradient descent.

$$L(W) = \frac{1}{2} \mathbb{E}[\|\hat{y} - y\|^2] \tag{5}$$

$$= \frac{1}{2} \mathbb{E}[\|\sum_{i=1}^{n} \sigma(W\bar{x}_i) - \sum_{i=1}^{n} \sigma(W^*\bar{x}_i)\|^2] \tag{6}$$

$$= \sum_{j=1}^{h} L_j(w_j) := \sum_{j=1}^{h} \frac{1}{2} \mathbb{E}[(\sum_{i=1}^{n} \sigma(w_j^\top \bar{x}_i) - \sum_{i=1}^{n} \sigma(w_j^{*\top} \bar{x}_i))^2]. \tag{7}$$

We will analyze the following gradient descent updates.

$$W_{t+1} = W_t - \eta \nabla L(W_t). \tag{8}$$

Here $\nabla L(W)$ denotes the gradient of the population loss and as is common in practice, we assume that the entries of the matrix $W$ are initialized i.i.d. from a Gaussian distribution. We are now ready to state our main theorem below.

**Theorem 1.** *Let $W^*$ be the unknown parameter for a one-round GNN in Eq. (4) with $\|w_j^*\| = 1$ for $j \in [h]$, and let $L(W)$ denote the population loss at $W$ as defined in Eq. (5). If the degree $d$ of the graph is $o(\sqrt{n})$, then for any $\epsilon \in (0, 1)$, if $W_0 \sim N(0, I)$, with probability at least $1 - h \cdot e^{-c'r}$ we have that $L(W_T) \leq \epsilon^2$ provided $T \geq c \cdot \frac{n^4 h^2 (d+1)}{\epsilon^2} \log(\frac{nh}{\epsilon})$ for absolute constants $c, c' > 0$.*

Several remarks are in order regarding the above theorem. Note that the convergence rate is $O(\frac{1}{\epsilon^2} \log(\frac{1}{\epsilon}))$ as a function of the desired error, and is polynomial in the other problem parameters such as the number of nodes, the maximum degree of the graph, and the dimensionality of the $W^*$ matrix. In the context of feedforward neural networks, gradient descent based algorithms have been analyzed primarily in the Neural Tangent Kernel (NTK) regime where an unknown target is learned by performing gradient descent on a highly over-parameterized network [Jacot et al., 2018,

Du et al., 2018, Daniely, 2017, Allen-Zhu et al., 2019]. The advantage of such an analysis is that it can be carried out for large depth neural networks (at least for smooth activations). However, due to the massive overparameterization the gradient descent updates move extremely slowly and the dynamics correspond to performing kernel regression in a very high dimensional space [Lee et al., 2019b, Arora et al., 2019b]. This is quite far from the behavior of gradient descent in realistic settings. We, on the other hand are interested in the setting where there is an unknown network generating the data and one would like to learn it via gradient descent with only mild over-parameterization. In particular, even the simple problem setting we consider in this section captures important aspects of the complexity of the general problem, in particular allowing for multiple local minima.

There have been recent efforts in going beyond the NTK regime for the case of feed-forward networks. Similar to our setting, these works assume that the input is generated from the Gaussian distribution. Furthermore, these works still either need some over-parameterization to argue convergence [Li et al., 2020], or need to add appropriate regularizers to the squared loss objective [Ge et al., 2017].

In contrast, an interesting aspect of our result is that we do not need any additional over-parameterization and learn the unknown GNN using another network of the same size. The recent work of Zhang et al. [2020] studied a model for GNNs similar to our setting in Eq. (4) and designed a learning algorithm that first initializes the network weights based on a tensor decomposition subroutine, followed by an accelerated gradient descent procedure. We on the other hand do not perform any special initialization based on techniques such as spectral methods and analyze gradient descent updates starting from Gaussian initialization. The model of GNN we consider in Eq. (4) also bears similarity to the model of overlapping patches considered in the work of Du et al. [2017] in the context of convolutional neural networks. The authors analyze the convergence of gradient descent under certain strong assumptions on the correlation among the different patch distributions. We on the other hand make no such assumptions. Furthermore, even in the case of feed-forward networks, current analysis in the non-NTK regime do not extend beyond one hidden layer. Hence our work brings the understanding of gradient descent for GNNs on par with that of feed-forward networks.

We next present the main ideas and key lemmas behind the proof of Theorem 1.

## 3.1 Expressions for loss and gradients

In this section we first compute simplified expressions for the loss and the gradients that will be useful in subsequent analysis. We first begin by writing an equivalent expression for the population loss. We have

$$L_j(w_j) = \frac{1}{2} \mathbb{E}\left[\left(\sum_{i=1}^{n} \sigma(w_j^\top \bar{x}_i)\right)^2\right] + \frac{1}{2} \mathbb{E}\left[\left(\sum_{i=1}^{n} \sigma(w_j^{*\top} \bar{x}_i)\right)^2\right] - \mathbb{E}\left[\sum_{i,j=1}^{n} \sigma(w_j^\top \bar{x}_i)\sigma(w_j^{*\top} \bar{x}_j)\right].$$

Notice that each $w_j$ evolves independently and hence we can simply focus on the convergence of $L_j(w_j)$ to $L_j(w_j^*)$. To simplify the above expression we will make use of the dual activation function of $\sigma(\cdot)$ from the work of Daniely et al. [2016] re-stated in Definition 1. For simplicity in this section we assume that the degree of each node is exactly $d$. The case when degree is at most $d$ follows with minimal changes and is presented in Appendix B. Notice that each $\bar{x}_i$ is a sum of $d+1$ messages, one involving the node $i$ and the other $d$ messages involving neighbors of $i$. Hence, $\bar{x}_i$ is a random variable distributed as $N(0, (d+1)I)$. Furthermore, for any $i \neq j$ define $d_{i,j}$ to be the number of common messages between $\bar{x}_i$ and $\bar{x}_j$. If $i = j$, then we define $d_{i,j} = d_{i,i} = d+1$. We next compute expressions for each of the three terms above.

$$\mathbb{E}\left[\left(\sum_{i=1}^{n} \sigma(w_j^\top \bar{x}_i)\right)^2\right] = \mathbb{E}\left[\sum_{i,j=1}^{n} \sigma(w_j^\top \bar{x}_i))\sigma(w_j^\top \bar{x}_j)\right]$$

$$= \mathbb{E}\left[(d+1)\|w_j\|^2 \sum_{i,j=1}^{n} \sigma\left(\frac{w_j^\top \bar{x}_i}{\sqrt{(d+1)}\|w_j\|}\right)\sigma\left(\frac{w_j^\top \bar{x}_j}{\sqrt{(d+1)}\|w_j\|}\right)\right]$$

$$= (d+1)\|w_j\|^2 \sum_{i,j=1}^{n} \hat{\sigma}\left(\frac{d_{i,j}}{d+1}\right) \quad \text{(definition of the dual activation in Eq. (3))}.$$

(9)

Similarly we get that,

$$\mathbb{E}\left[(\sum_{i=1}^{n}\sigma(w_j^{*\top}\bar{x}_i))^2\right] = (d+1)\|w_j^*\|^2\sum_{i,j=1}^{n}\hat{\sigma}(\frac{d_{i,j}}{d+1}). \tag{10}$$

Finally we simplify the last term.

$$\mathbb{E}\left[\sum_{i,j=1}^{n}\sigma(w_j^\top\bar{x}_i)\sigma(w_j^{*\top}\bar{x}_j)\right] = \mathbb{E}\left[(d+1)\|w_j\|\|w_j^*\|\sum_{i,j=1}^{n}\sigma(\frac{w_j^\top\bar{x}_i}{\sqrt{(d+1)}\|w_j\|})\sigma(\frac{w_j^{*\top}\bar{x}_j}{\sqrt{(d+1)}\|w_j^*\|})\right]$$

$$= (d+1)\|w_j\|\|w_j^*\|\sum_{i,j=1}^{n}\hat{\sigma}(\frac{d_{i,j}}{d+1}\frac{w_j^\top w_j^*}{\|w_j\|\|w_j^*\|}). \tag{11}$$

Combining the (9), (10), and (11), we have

$$L(w_j) = \frac{1}{2}(d+1)\|w_j\|^2\sum_{i,j=1}^{n}\hat{\sigma}(\frac{d_{i,j}}{d+1}) + \frac{1}{2}(d+1)\|w_j^*\|^2\sum_{i,j=1}^{n}\hat{\sigma}(\frac{d_{i,j}}{d+1})$$

$$- (d+1)\|w_j\|\|w_j^*\|\sum_{i,j=1}^{n}\hat{\sigma}(\frac{d_{i,j}}{d+1}\frac{w_j^\top w_j^*}{\|w_j\|\|w_j^*\|}). \tag{12}$$

It is easy to see that if $w_{0,j}$ is the initial value of $w_j$ then each subsequent iteration will be a linear combination of $w_{0,j}$ and $w_j^*$. Hence we can assume that $w_j = \alpha w_j^* + \beta w_j^\perp$, where $w_j^\perp$ is a fixed unit vector (depending on the initialization) orthogonal to $w_j^*$. Then re-writing the loss in terms of $\alpha, \beta$ and recalling that $\|w_j^*\| = 1$ we get the simplified expression:

$$L(\alpha,\beta) = \frac{1}{2}(d+1)(\alpha^2+\beta^2)\sum_{i,j=1}^{n}\hat{\sigma}(\frac{d_{i,j}}{d+1}) + \frac{1}{2}(d+1)\sum_{i,j=1}^{n}\hat{\sigma}(\frac{d_{i,j}}{d+1})$$

$$- (d+1)\sqrt{\alpha^2+\beta^2}\sum_{i,j=1}^{n}\hat{\sigma}(\frac{d_{i,j}}{d+1}\frac{\alpha}{\sqrt{\alpha^2+\beta^2}}). \tag{13}$$

In the rest of the section we will analyze the evolution of the updates of $\alpha$ and $\beta$. Furthermore we will use $\alpha_t, \beta_t$ to denote the parameters $\alpha, \beta$ associated with the iterate $w_{j,t}$ at time $t$. We first compute the gradient of the objective w.r.t. $w$ or equivalently w.r.t. $\alpha, \beta$.

$$\frac{\partial L(\alpha,\beta)}{\partial\alpha} = \alpha(d+1)\sum_{i,j=1}^{n}\hat{\sigma}(\frac{d_{i,j}}{d+1}) - \frac{\alpha}{\sqrt{\alpha^2+\beta^2}}(d+1)\sum_{i,j=1}^{n}\hat{\sigma}(\frac{d_{i,j}}{d+1}\frac{\alpha}{\sqrt{\alpha^2+\beta^2}})$$

$$- \frac{\beta^2}{\alpha^2+\beta^2}(d+1)\sum_{i,j=1}^{n}\frac{d_{i,j}}{d+1}\hat{\sigma}'(\frac{d_{i,j}}{d+1}\frac{\alpha}{\sqrt{\alpha^2+\beta^2}})$$

$$= \alpha(d+1)\sum_{i,j=1}^{n}\hat{\sigma}(\frac{d_{i,j}}{d+1}) - \frac{\alpha}{\sqrt{\alpha^2+\beta^2}}(d+1)\sum_{i,j=1}^{n}\hat{\sigma}(\frac{d_{i,j}}{d+1}\frac{\alpha}{\sqrt{\alpha^2+\beta^2}})$$

$$- \frac{\beta^2}{\alpha^2+\beta^2}(d+1)\sum_{i,j=1}^{n}\frac{d_{i,j}}{d+1}\hat{\sigma}_{\text{step}}(\frac{d_{i,j}}{d+1}\frac{\alpha}{\sqrt{\alpha^2+\beta^2}}). \tag{14}$$

Here in the last equality we have used the fact that $\hat{\sigma}' = \hat{\sigma'}$ and that $\sigma'(x) = \sqrt{2}\mathbb{1}(x\geq 0) = \sigma_{\text{step}}(x)$, where $\sigma_{\text{step}}(x)$ is the step function. See [Daniely et al., 2016] for a proof. Similarly we get that

$$\frac{\partial L(\alpha,\beta)}{\partial\beta} = \beta(d+1)\sum_{i,j=1}^{n}\hat{\sigma}(\frac{d_{i,j}}{d+1}) - \frac{\beta}{\sqrt{\alpha^2+\beta^2}}(d+1)\sum_{i,j=1}^{n}\hat{\sigma}(\frac{d_{i,j}}{d+1}\frac{\alpha}{\sqrt{\alpha^2+\beta^2}})$$

$$+ \frac{\alpha\beta}{\alpha^2+\beta^2}(d+1)\sum_{i,j=1}^{n}\frac{d_{i,j}}{d+1}\hat{\sigma}_{\text{step}}(\frac{d_{i,j}}{d+1}\frac{\alpha}{\sqrt{\alpha^2+\beta^2}}). \tag{15}$$

## 3.2 Bounding the iterates.

In this section we show that if the initial weight $w_{0,j}$ is drawn from $N(0, \sigma^2 I)$ then with high probability, the iterates remain bounded for all subsequent time steps. We first analyze how the length of $w$, i.e., $\|w_t\|^2$ behaves over a period of time. In the rest of the proof we will often use the shorthand $A, B_t, C_t$ to denote the following key quantities that depend on the structure of the graph.

$$A = \sum_{i,j=1}^{n} \hat{\sigma}\left(\frac{d_{i,j}}{d+1}\right)$$

$$B_t = \sum_{i,j=1}^{n} \hat{\sigma}\left(\frac{d_{i,j}}{d+1} \frac{\alpha_t}{\ell_t}\right)$$

$$C_t = \sum_{i,j=1}^{n} \frac{d_{i,j}}{d+1} \hat{\sigma}_{\text{step}}\left(\frac{d_{i,j}}{d+1} \frac{\alpha_t}{\ell_t}\right).$$

Here $\ell_t$ denotes the length of the iterate $w_t$ at time $t$, i.e., $\ell_t = \|w\|_t = \sqrt{\alpha_t^2 + \beta_t^2}$. The above quantities satisfy useful inequalities that will be used throughout the analysis. For example, we note that $A, B_t$ are always non-negative when $d = o(\sqrt{n})$ and $\alpha_t \geq 0$. We also have $\frac{n^2}{4\pi} \leq |B_t| \leq A$ and $|C_t| = o(A)$. See Appendix B for the proof.

**Lemma 1.** *If* $w_{0,j} \sim N(0, \sigma^2 I)$ *and* $\eta \in \left[\frac{1}{16\pi(d+1)A}, \frac{1}{6\pi(d+1)A}\right]$ *then with probability at least* $1 - e^{-O(r)}$*, it holds that for all* $t \geq 0$*,* $\ell_t \leq 2\sigma\sqrt{r} + 4\pi + 1$*, and* $\ell_t \geq \frac{1}{36\pi^2}$ *for all* $t \geq 1$*.*

## 3.3 Establishing smoothness and analyzing initial updates

We first show that if the condition in Lemma 1 holds, then throughout the trajectory of the iterates, the loss function is smooth. This is formalized in the lemma below.

**Lemma 2.** *If the degree* $d$ *of the graph is* $o(\sqrt{n})$ *and the conditions in Lemma 1 hold, then for all* $t \geq 1$ *and any* $\gamma \in [0, 1]$ *such that* $(\alpha, \beta) = (1 - \gamma)(\alpha_t, \beta_t) + \gamma(\alpha_{t+1}, \beta_{t+1})$*, we have that*

$$\lambda_{max}(\nabla^2 L(\alpha, \beta)) \leq 4(d+1)A\left(1 + \sqrt{2\sigma\sqrt{r} + 4\pi + 1} + o(1)\right).$$

Here $\lambda_{\max}$ is the maximum eigenvalue of the population Hessian denoted by $\nabla^2 L(\alpha, \beta)$. Our overall proof strategy is to show that the Polyak-Łojasiewicz (PL) condition [Polyak, 1963] holds, namely that the squared norm of the gradient lower bounds the loss value at any iterate. This will let us easily analyze convergence of gradient descent via standard arguments. However a challenge is that since we are starting from random initialization, the PL condition does not hold true at the beginning. As a result we separately analyze an initial phase of the algorithm and show that gradient descent very quickly escapes a region where the PL condition does not hold. This is formalized below.

**Lemma 3.** *If* $w_{0,j} \sim N(0, \sigma^2 I)$ *and* $\eta \in \left[\frac{1}{16\pi(d+1)A}, \frac{1}{6\pi(d+1)A}\right]$ *then with at least* $1 - 1/h^2$*, it holds that for all* $t \geq c \log(\sigma \log h)$*,* $\alpha_t \geq -\frac{1}{100}$ *and* $\ell_t \geq 1 - o(1)$*, where* $c > 0$ *is an absolute constant.*

## 3.4 Establishing the PL-condition

Finally, using the above lemma we show that under the conditions of Lemma 1 the iterates $w_t = (\alpha_t, \beta_t)$ satisfy the Polyak-Łojasiewicz inequality [Polyak, 1963].

**Lemma 4.** *If the degree* $d$ *of the graph is* $o(\sqrt{n})$ *and the conditions in Lemma 1 hold then there is an absolute constant* $c > 0$*, such that for all* $t \geq c \log(\sigma \log h)$ *and* $\epsilon \in (0, 1)$*, either* $|\beta_t| \leq \frac{\epsilon}{4hn}$ *and* $\|\ell_t - 1\| \leq \frac{\epsilon}{4hn}$ *or we have that*

$$\|\nabla L(\alpha_t, \beta_t)\|^2 \geq \mu^* L(\alpha_t, \beta_t),$$

*where* $\mu^* \geq \frac{\epsilon^2}{380(d+1)h^2\pi n^2}$*.*

## 3.5 Putting everything together

Finally, we combine the analysis from the previous sections to prove the main theorem, i.e., Theorem 1.

*Proof of Theorem 1.* We will analyze an arbitrary $j \in [h]$ and the evolution of the corresponding $w_j$ vector. By setting $\sigma = 1$ we have from Lemma 2 that the smoothness parameter of the loss function is

$$L \leq 4(d+1)A\Big(1 + \sqrt{4\pi + 3} + o(1)\Big).$$

Hence we get that for any $t \geq 0$,

$$
\begin{aligned}
L_j(w_{j,t+1}) &\leq L_j(w_{j,t}) + \nabla L_j(w_{j,t})(w_{j,t+1} - w_{j,t}) + \frac{L}{2}\|w_{j,t+1} - w_{j,t}\|^2 \\
&\leq L_j(w_{j,t}) - \eta\|\nabla L_j(w_{j,t})\|^2 + \frac{\eta^2 L}{2}\|\nabla L_j(w_{j,t})\|^2 \\
&= L_j(w_{j,t}) - \eta\|\nabla L_j(w_{j,t})\|^2(1 - \frac{\eta L}{2}).
\end{aligned}
\tag{16}
$$

From the range of $\eta$ in Lemma 1 we get that $\eta L \leq 1$. Furthermore, using Lemma 4 we can write

$$
\begin{aligned}
L_j(w_{j,t+1}) &\leq L_j(w_{j,t}) + \nabla L_j(w_{j,t})(w_{j,t+1} - w_{j,t}) + \frac{L}{2}\|w_{j,t+1} - w_{j,t}\|^2 \\
&\leq L_j(w_{j,t})(1 - \eta\mu^*) \\
&\leq L_j(w_{j,0})(1 - \eta\mu^*)^t.
\end{aligned}
\tag{17}
$$

Hence after $T \geq c \cdot \frac{n^4 h^2 (d+1)}{\epsilon^2}\log(\frac{nh}{\epsilon})$ time steps we will either have $L_j(w_{j,t}) \leq \epsilon^2/h$, or that $|\beta_t| \leq \frac{\epsilon}{4hn}$ and $\|\ell_t - 1\| \leq \frac{\epsilon}{4hn}$. The latter implies that

$$\|w_{j,t} - w_j^*\|^2 \leq \frac{\epsilon^2}{2hn^2}.$$

Furthermore, it is easy to see that

$$L_j(w_{j,t}) \leq n(n-1)\|w_{j,t} - w_j^*\|^2.$$

Hence if the latter happens then again, $L_j(w_{j,t}) \leq \epsilon^2/h$ thereby implying that $L(W_T) \leq \epsilon^2$. $\square$

## 4 Deep linear GNNs

In the previous section we analyzed one round GNNs with ReLU activations. Analyzing the dynamics of gradient descent beyond one layer (in the non NTK regime) with non-linear activations is a challenging problem, even for standard network architectures. Hence, in this section we consider deep linear GNNs in order to analyze the dynamics of gradient descent in deeper message passing architectures. Linearity has been a commonly studied setting in recent years for analyzing deep feed-forward networks [Hardt and Ma, 2016, Bartlett et al., 2018, Arora et al., 2018a].

We assume the existence of an unknown $L$-round GNN characterized by positive-definite symmetric matrices $W_1^*, W_2^* \in \mathbb{R}^{r \times r}$. Given node inputs $x_1, \ldots, x_n \sim N(0, I_{r \times r})$ the embedding of node $i$ at round $\ell \geq 1$ is defined as

$$a_i^{(\ell)} = \text{AGGREGATE}(\{x_j^{\ell-1} : j \in N(i)\}) = W_2^* \sum_{j \in N(i)} x_j^{(\ell-1)}. \tag{18}$$

$$x_i^{(\ell)} = \text{COMBINE}(x_i^{(\ell-1)}, a_i^\ell) = W_1^* x_i^{(\ell-1)} + a_i^{(\ell)}. \tag{19}$$

Notice that this is a more general setting than the case of ReLU network studied in the previous section where the aggregate was a simple summation operation. In this section we can analyze the case where both the aggregate and combine operations have their own set of parameters $W_1^*$ and $W_2^*$. Finally, the output of the network on input $\vec{x} = \{x_1, \ldots, x_n\}$ is defined as

$$y = \sum_{i=1}^{n} x_i^{(L)}.$$

As in the case of ReLU activations from the previous section, we will analyze the non-overparameterized case where another architecture of the same size is used to learn the unknown ground truth network. Given another $L$-round GNN defined by parameters $W_1, W_2$ we define the population loss as

$$L(W_1, W_2) = \frac{1}{2} \mathbb{E}\left[\|(\hat{y} - y)\|^2\right] \tag{20}$$

$$= \frac{1}{2} \mathbb{E}\left[\|(\sum_{i=1}^{n} \hat{x}_i^{(L)} - \sum_{i=1}^{n} x_i^{(L)})\|^2\right]. \tag{21}$$

We will analyze the gradient descent updates defined as

$$W_{1,t+1} = W_{1,t} - \eta \nabla L(W_{1,t}, W_{2,t}) \tag{22}$$
$$W_{2,t+1} = W_{2,t} - \eta \nabla L(W_{1,t}, W_{2,t}). \tag{23}$$

For ease of interpretation, we state our main theorem below for the case when each node in the graph has degree exactly $d$. The general theorem is stated in Appendix C and achieves similar rate of convergence. The theorem below shows that if the network parameters are initialized at identity, i.e., $W_{1,0}, W_{2,0} = I$, then gradient descent minimizes the population loss at a polylogartihmic rate.

**Theorem 2.** *Let the initialization of $W_{1,0}, W_{2,0}$ be identity. Then, $L(W_{1,T}, W_{2,T}) \leq \epsilon^2$ if,*

$$T \geq \max_{i \in [n]} \frac{1}{\eta_0 L^2 (d^2 + 1) \ell_i^{2L-2}} \log\left(\frac{rLu_i((1 - \sigma_i^*)^2)}{\epsilon}\right) \tag{24}$$

$$\eta := \eta_0 \leq \min_i \frac{\ell_i^{2L-1} \min(1, \frac{1}{(\sigma_i^*)^{2L-2}})}{2nL^2(\delta_i)^2(d^2 + 1)u_i^{2L-3}(1 + u_i)^{L-1}}. \tag{25}$$

*Here $\sigma_i^*$ is the $i$th smallest singular value of $W_1^* + dW_2^*$, and $u_i$, $\ell_i$ and $\delta_i$ depend only on $\sigma_i^*$.*

We make a few comments about the theorem above. Notice that unlike the case of ReLU networks we do not assume any bound on the maximum degree of the graph and the theorem applies generally. Furthermore, the convergence rate is logarithmic in $\frac{1}{\epsilon}$ and exponential in the depth $L$ of the network. The dependence on the number of nodes in the graph is linear and the dependence on the maximum degree $d$ is polynomial and appears via the bound on the maximum singular value of the matrix $W_1^* + dW_2^*$. Our analysis for the case of equal degrees proceeds by showing that it is enough to track the evolution of the singular values $\sigma_{i,t}$ of the matrix $M_t = W_{1,t} + dW_{2,t}$ at each time step. We then show that is $\eta$ is sufficiently small then the singular values $\sigma_{i,t}$ will converge to the singular values $\sigma_i^*$ of the true unknown matrix $W_1^* + dW_2^*$. Handling the case of unequal degrees is along similar lines but is technically more challenging as we need to separately track the evolution of the singular values of $W_{1,t}$ and $W_{2,t}$ separately. See Appendix C for details.

## 5 Experiments

In this section we verify our theoretical results via simulations. Our first goal is to understand whether the $\sim \frac{1}{\epsilon^2}$ rate of convergence obtained for the case of one round ReLU GNNs is tight, or whether in practice we can obtain polylog($\frac{1}{\epsilon}$) rates even for the ReLU setting. Secondly, we understand via experiments the regime where the degree of the graph $d$ exceeds $\sqrt{n}$, and hence our theoretical results for the case of ReLU GNNs do not hold. In order to do this we generate random $d$-regular graphs over $n = 100$ nodes. For the case of one round GNNs with ReLU activations we set the embedding size $r = 10$, and $h = 10$ (number of hidden units in the ReLU GNN). We use the same value of $r$ for the case of deep linear GNNs, where $r$ equals the input dimensionality and also the dimensionality of the matrices $W_1^*$, and $W_2^*$.

We generate an unknown ground truth network in the case of ReLU GNNs by choosing each column of $W^*$ to be a random unit length vector. For the case of linear networks we generate positive definite matrices $W_1^*, W_2^*$ by picking random Gaussian entries, and then adding a small multiplicative factor of 0.001 times the identity matrix. As dictated by our theory, we initialize network weights from a standard Gaussian for the case of ReLU networks, and to the identity matrix for the linear case.

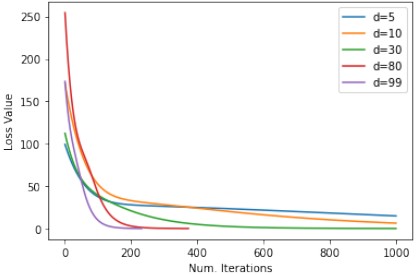

Figure 1: Loss vs. number of iterations of gradient descent for one round GNN with ReLU activations.

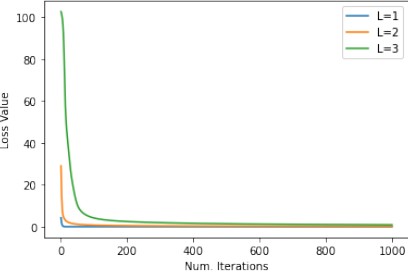

Figure 2: Loss vs. number of iterations of gradient descent for deep linear GNNs.

We simulate population gradient descent and implement our networks using the JAX programming language [Bradbury et al., 2018]. Our experiments are run using one GPU. See Section D for further details.

For the case of ReLU networks we run gradient descent on instances with varying values of the degree $d$. In particular we present results for $d \in \{5, 10, 30, 80, 99\}$. For the case of linear networks we vary the depth $L$ in $\{1, 2, 3\}$. Finally, in each case we plot the loss value ($\ell_2$ loss) vs. the number of iterations. The results are shown in Figure 1 and Figure 2. We make a few remarks about the experiments. Note that the convergence rate for the case of linear networks is indeed much better than the case of ReLU GNNs and hence we, in general, do not expect significantly better rates of convergence than the $\sim \frac{1}{\epsilon^2}$ bound that we prove. An interesting observation is that as $d$ increases and is close to the regime of $\tilde{\Theta}(n)$, the rates of convergence of gradient descent for the case of a one round GNN with ReLU activations, become much better. This is along expected lines, since if $d = n - 1$, i.e, the graph is a complete graph, then the network in Eq. (4) boils down to $h$ independent single ReLU units with standard Gaussian inputs. It is known for this extreme setting that gradient descent has a logarithmic convergence in $\frac{1}{\epsilon}$ [Soltanolkotabi, 2017]. Hence, extending our existing theoretical analysis beyond the $\sqrt{n}$ degree setting is an interesting open question.

## 6   Conclusions

The main question left open by our work is to analyze the gradient descent updates for multi-round GNNs with ReLU activations. Given the difficulty of analyzing gradient descent for multi-layer networks beyond the NTK setting, the above seems like a challenging problem. It would also be interesting to explore whether the convergence rate for the case of one round GNNs can be improved via over-parameterization. Finally, it would also be of interest to extend our results to more general distributions going beyond the Gaussian setting. Another fascinating question is to analyze gradient descent in the *agnostic* setup. If the best one round GNN with ReLU activations (or a deep linear GNN) achieves an error of OPT on the data distribution, can gradient descent recover a network of error $f(\text{OPT})$? This style of analysis has recently been carried out in the case of a single ReLU unit [Frei et al., 2020].

## Acknowledgments and Disclosure of Funding

Funding in direct support of this work: Google Research.

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
