# OpenReview forum: "A Convergence Analysis of Gradient Descent on Graph Neural Networks"
_NeurIPS.cc/2021/Conference — NeurIPS 2021 Poster_

### Official Review · Reviewer_iYkN · 2021-07-05

**Rating:** 6
**Confidence:** 4

**Summary:**

In this paper, the authors analyze the convergence of gradient descent on message-passing Graph Neural Networks in the realisable case. They consider two settings: one-layer GNN with ReLU, and deep, linear, recurrent GNNs. They show that, if the input node features are drawn from iid Gaussians and the graph is sparse enough (degrees are bounded by the square root of the number of nodes), then minimizing the expected square loss will lead to convergence toward the true weights of the GNN. Explicit rate are given with respect to the various parameters of the problem.

**Limitations And Societal Impact:**

See above for limitations, which is my main comment.

This is a very theoretical work.

**Main Review:**

This is an interesting paper, that extends to (simple) GNNs a now well-furnished line of works on the convergence of deep NNs in the realizable case. The paper is well-written and includes a sketch of proof. The proof method is (as far as I know) rather classic, the graph only appearing mostly through its degrees, but some computations are non-trivial.

My main comment is that, while the authors make an appreciable effort to discuss the simplifications made for this analysis, the most striking one is not discussed: unless I am missing something, the authors minimize the *expected* risk, with the expectation taken with respect to the random node features. At first glance, I was wondering why the authors don't extend the analysis to the minimization of the empirical risk as is usually done, but in fact this is here highly unrealistic: it would assume that, for each sample graph, the graph structure is the same ! This deserves at least a highly emphasized remark in the paper.
A potential solution would be to "create" an empirical risk by dividing the coordinates of the Gaussian node features into subgroups, create each sample graph with a different subgroup of coordinates, and average over them. This would effectively correspond to having several copies of the same graph structure but with several independent node features, which would allow approaching the expected risk. Do the authors think that this could be feasible, via classical concentration arguments ?
Another possibility would be to assume that, for each sample, the graph structure itself depend on the node features like in the random graph literature (see eg the works of Ruiz, Ribeiro et al, or Keriven, Bietti and Vaiter for links between GNN and random graphs) but the analysis seems fairly different in this case, and probably significantly more involved.
With this in mind, the numerical experiments do not make much sense. The minimized risk involves only one graph sample, and is probably very far from the expected risk. Is there an argument that allows to go from one to the other ?

Other comments/typos:
- The deep linear GNN seems to have the same weights at each layer, being effectively a "recurrent" network. Is there a reason for this ?
- in the sketch of proof of the one layer case, additional efforts could be dedicated to "high-level" explanations of the proof rather than hand-on computations. After all, the GNN is just $ sigma(W*A*X) $, with adjacency matrix A and iid Gaussians X, and the graph structure appears only through the d_ij (which are just the entries of A^2).
- in section 3.1 (at least), the index j is used both for the weight and as a summation index, while it should be a different index. This is very confusing.
- two spurious parenthesis in the first line of eq (9).

To conclude, this paper has merits, but a very significant blind spot on the expected/empirical risk aspect. My note may change quite a lot after discussion.

**===== Edit after rebuttal =====**

I thank the authors for their clear response. Although the "fixed graph" settings still remain a big limitation of their work, they acknowledge it and put forth convincing arguments. I am confident that they will clarify these limitations in the paper, as well as emphasize future directions to alleviate them. I thus believe that this paper might be an interesting first step. I increased my score.

**Time Spent Reviewing:**

3

---

> ### Author Response · Authors · 2021-08-10
> **Response to reviewer iYkN**
>
> Thank for a very insightful review. Below we address specific technical questions.
>
> **Q: Empirical vs. Expected Risk:**
>
> **Response:** Thank you for asking this question. There are two natural settings where the “empirical vs. expected” transformation can be easily carried out. One setting is when the graph structure $G$ is fixed and the i.i.d. samples correspond to the sampling of the node features. This scenario captures many practical settings such as applying GNNs to road networks (where the graph structure is fixed). In this case eq. (5) is the expected risk and standard uniform convergence bound will apply. Note that the experiments were performed with this setting in mind. In fact, in the experiments we perform exact gradient descent on the objective in eq. (5).
>
> The other natural setting is where the i.i.d. samples include both a sampled graph and sampled node features. For instance consider the case where the graph $G$ is sampled from a distribution over graphs with degree at most $d$, and conditioned on $G$, the node features are sampled as described in the paper. Then again uniform standard convergence bounds will apply and it is enough to analyze the expected risk. Hence eq. (5) will have another expectation over the sampling of $G$. In this case all our analysis goes through by re-defining the quantities $A$,$B_t$ and $C_t$ with their expectations (over the sampling of the graph). Notice that our analysis relies on certain relationships between $A$, $B_t$, and $C_t$ that hold “pointwise” for each degree at most $d$ graph (see section 3.2), and hence will hold in expectation as well. Hence our analysis of gradient descent still holds in this setting. We chose to highlight the simple form of the risk in eq. (5) since that captures the main complexity of analyzing gradient descent updates in the above two settings. We are happy to include this in the paper for a more complete discussion.
>
> The most challenging case is when the graph $G$ and the node features are sampled in a correlated manner, in which case the uniform convergence bounds can still be carried out, but our analysis of the gradient descent updates on the expected risk will not hold. However, this is quite a challenging setting. At the very least this would require analyzing gradient descent updates in the non-NTK/mean-field regime (i.e. infinite/large width regime) without the i.i.d. Gaussianity assumption.  We do not know of any existing analysis of gradient descent, for any class of architectures, that can deal with such correlations.
>
> We are more than happy to engage further and provide more clarifications.
>
> **Q: The deep linear GNN seems to have the same weights at each layer, being effectively a "recurrent" network. Is there a reason for this ?**
>
> **Response:** There are existing variants of GNN that incorporate weight sharing across layers (for example, https://arxiv.org/abs/1511.05493). Furthermore, since we are analyzing a deep network, the analysis is already quite non-trivial in this setting and we believe that our simplified deep linear model is a good starting point towards building a more general theoretical analysis. We believe that our techniques can be extended to the case when the weights are not shared. This is a direction for future work.

---

### Official Review · Reviewer_jXXp · 2021-07-08

**Rating:** 6
**Confidence:** 3

**Summary:**

This paper analyzes the convergence of gradient descent (GD) on graph neural networks (GNNs). More specifically, the authors derive convergence rates for GD on a 1-layer GNNs with ReLU activation function, and on a multi-layer GNN without nonlinearities.

**Main Review:**

Originality: This paper presents an original convergence analysis for GD on GNNs. The analysis builds on a similar convergence analysis developed for fully connected neural networks.

Quality/Significance:  While the contribution has some significance and the analysis seems to be technically sound, the quality of the paper is limited by the fact that the authors don't discuss the limitations of the models they consider. I also find it strange that they do not discuss convexity. The first limitation that I see is that the analyzed models are not very useful in practice. Multi-layer GNNs without nonlinearities are just linear (over)parametrizations that could be written much more simply as one linear layer, and 1-layer ReLU GNNs are too small for most applications. I understand that analyzing multi-layer GNNs with nonlinearities is not trivial, but the authors should have better stressed the limitations of their approach. I also think it important to mention that the ERM problem with quadratic loss is convex for the 1-layer ReLU GNN. This alone implies convergence of gradient descent.

Clarity: The writing is clear but the organization of the paper could be improved. The authors discuss related work in Section 2, but also include some of it in Section 3 where they put more emphasis on contrasting their results with prior work than on discussing Theorem 1.  The content of Sections 3 and 4, which contain the paper's contributions (the convergence analyses for the 1-layer and multi-layer GNNs respectively), is extremely imbalanced: Section 3 is long and contains too many technical details, reading more like a proof than a paper; in contrast, Section 4 is short and looks rushed. This would be expected if the analysis is Section 4 were an extension of the analysis of Section 3, but this is not the case. I also wish the authors had discussed the Polyak condition in more detail; that would have been helpful for a high-level understanding of the proof strategy. Finally, there is a slight problem with notation---\sigma is used in three different places to mean three different things---and a few typos here and there (e.g., lines 90 and 116; lowercase o for big oh notation in line 115).

*** Update after rebuttal ***

Thank you for the clarification, indeed convexity would only hold if the labels were zero or negative. Subject to a more detailed discussion on the limitations of the models considered and on improvements in the organization of the paper (as per my original review), I am increasing my score to a 6.

**Time Spent Reviewing:**

3h

---

> ### Author Response · Authors · 2021-08-10
> **Response to reviewer jXXp**
>
> Thank you for your review. Below we address specific technical questions.
>
> **Q: On Convexity:**
>
> **Response:** We would like to clarify an important mis-understanding. Our loss function is not convex (as a function of the network parameters). In fact even in the case of a single ReLU network (a graph with one node) and squared loss the objective in eq. (5) is non-convex (in the parameters). This was already observed in prior works (see https://arxiv.org/abs/1705.04591). Below is a simple example to demonstrate non-convexity.
>
> Consider a graph with just one node. In this case the squared loss becomes equal to
>
> $L(w) = E [\sigma(w \cdot x) - \sigma(w^* \cdot x)]^2$.
>
> Now consider three points ($w_1 = \epsilon w^*$, $w_2 = -\epsilon w^*$, $w_3 = 0$) where $\epsilon < 1$. Then it is easy to check that
>
> $L(w_1) = \frac{\epsilon^2}{2} + \frac{1}{2} - \epsilon$, and
> $L(w_2) = \frac{1}{2} + \frac{\epsilon^2}{2}$.
>
> However, $L(0) = \frac{1}{2} > \frac{1}{2} (L(w_1) + L(w_2))$.
>
> Furthermore, in the case of more than one node, it is not even the case that any local optimum is a global optimum. If that were the case gradient descent from any initialization would converge. However, in our setting random initialization plays a crucial role. We first need to show that via random initialization, the starting point will be not too negatively correlated with $w^*$ and will after a few steps enter a region where the PL condition can be satisfied (although the loss may again be non-convex in this region). This requires quite non-trivial analysis.
>
> We sincerely hope that you will re-evaluate our paper based on the above. We also appreciate your suggestions on improving the writing and discussion of related work in the main body. We will certainly incorporate them.
>
> **Q: The first limitation that I see is that the analyzed models are not very useful in practice. Multi-layer GNNs without nonlinearities are just linear (over)parametrizations that could be written much more simply as one linear layer, and 1-layer ReLU GNNs are too small for most applications.**
>
>
> **Response:** We would like to point out that understanding and analyzing gradient descent for optimization and generalization in neural networks is a very active area of research. In order to develop a general theory that applies to deep nonlinear networks one has to make strong assumptions such as the width going to infinity (NTK and mean-field regimes).
> Another line of research, which is the focus of our setting, is to work in the low width regime where one explicitly avoids making NTK type assumptions. In these scenarios all existing analyses of neural networks (these analyses mainly exist for fully connected networks) work either with nonlinear networks with one hidden layer or with deep linear networks (see references below). Hence our contribution through this work is to expand the latter line of research and bring the case of GNNs closer to the more well understood case of fully connected networks.  While none of the above types of analyses provide a complete answer to the properties of gradient descent on “realistic” neural networks, these are valuable theoretical contributions towards the extremely challenging goal of understanding gradient descent based optimization for neural networks.
>
> https://arxiv.org/abs/1810.02281
>
> https://arxiv.org/abs/1711.00501
>
> https://arxiv.org/abs/1705.09886
>
> https://arxiv.org/abs/2007.04596
>
> https://arxiv.org/pdf/1902.04674.pdf
>
> https://arxiv.org/abs/1802.06093

---

### Official Review · Reviewer_iGLS · 2021-07-16

**Rating:** 6
**Confidence:** 3

**Summary:**

This paper analyzed the optimization performance of two types of GNNs. Specifically, this paper derived the convergence rate to a global optimum with gradient descent in the realizable case for ReLU-GNNs with one hidden layer and deep linear-GNNs. This paper showed that the former achieves a linear rate, and the latter achieves the $\varepsilon$ error with $O(1/\varepsilon^2 \log(1/\varepsilon))$  iterations. Furthermore, this paper experimentally verified the behavior of loss function with respect to model parameters and hyperparameters such as the number of nodes and number of layers.

**Ethical Concerns:**

N.A.

**Limitations And Societal Impact:**

- [L1] This paper discussed the limitations of the analysis. Specifically, the analysis is only applicable to $d=o(\sqrt{n})$ (l.273.) Also, it is not applicable to non-linear deep GNNs (l.276.)
- [L2] See [W1] for the other possible limitations to be addressed.

**Main Review:**

### Strength

- [S1] To the best of my knowledge, this is the first paper to show the global convergence of non-linear GNNs at a linear rate by exploiting the realizability condition.
- [S2] The paper is well-written and easy to follow.

### Weakness

- [W1] The problem is setting is different from usual node prediction tasks (see the Soundness section).
- [W2] In addition, there are substantial restrictions on data distribution and GNN architecture. (For example, deep GNNs are linear and use weight-sharing.)
- [W3] I have some questions about the soundness of the experiment (see the Soundness section).

### Soundness (Do theorems and experiments answer research questions, assuming they are correct?)

- [So1] I think the problem setting considered in this paper is different from the usual node prediction tasks. Therefore, we cannot directly apply the results of this paper. Specifically, in this paper, the model can observe teacher signal y of all nodes. However, we usually cannot see the teacher signal (y) of test nodes in node prediction tasks. Is it possible to apply this method to the partially observed setting?
- [So2] The deep linear GNN used in this paper shares weights across layers. l.79 pointed out that GCN is such an architecture. But I think this is not true -- GCN uses different weights for each layer. It is OK to analyze the weight-sharing GNNs. However, I believe it is necessary to clarify that this architecture is different from the usual GCNs.
- [So3] I have a question about whether the discussion in the experiment section is appropriate. This paper wrote that one of the experiment's objectives is to see whether the rate of ReLU GNN is tight or not (l.246.) and concluded that it is practically tight. However, I think it is difficult to evaluate the rate quantitatively just by comparing the loss curves of deep linear GNNs and ReLU GNNs with a single hidden layer.
- [So4] Regarding the second objective of the experiment (behavior of GNNs in the regime of $d=O(n)$), I recognized its soundness.

### Correctness (Are derivation of theorems and experiments correct?)

- [C1] As far as I checked, there were no major errors in the proof.
- [C2] I have one question about the proof of Theorem 1. Lemma 4 showed that either (1) the conditions on $\beta_t$ and $\ell_t$ or (2) the condition on $\nabla L$ held for each $t$. However, it seems that the proof of Theorem 1 implicitly assumed that the condition (2) held for any $t\leq T$ up to a given $T$.

### Novelty and Significance (Do the paper have novel points? If so, are they significant?)

- [N1] The strategy of using the PL condition to show linear convergence is a standard one. However, the proof has novelty in proving the satisfiability of the PL condition.
- [N2] As far as I know, this is the first paper that showed the linear convergence of non-linear GNN to a global optimum under realizability conditions. Indeed, the architecture and data distribution of GNNs are different from those of real-world applications (e.g., realizable setting, random input, observability of teacher signals.) With that being said, I think this paper is significant as this paper is a good first step toward understanding the optimization perspective of GNNs.
- [N3] For GNN optimization, [Oono & Suzuki, NeurIPS2020] and [Xu et al., ICML2021] considered the optimization performance of GNNs. However, this paper has novelty compared with these papers. Specifically, these papers analyzed multi-scale GNNs, while this paper single-scale GNNs. In addition, existing papers made assumptions other than realizability (weak-learning condition for [Oono & Suzuki, NeurIPS2020], the positivity of the (submatrix of) adjacency matrix for [Xu et al., ICML2021].)
- [N4] This paper claimed that there is no convergence analysis of gradient descent for feed-forward NNs with more than two hidden layers in the non-NTK regime (l.146.) However, [Pham & Nguyen, ICLR2021] analyzed of NNs with two hidden layers in the mean-field regime. Therefore, the claim does not seem correct, judging solely from this sentence.
- [Oono and Suzuki, NeurIPS2020] https://proceedings.neurips.cc/paper/2020/hash/dab49080d80c724aad5ebf158d63df41-Abstract.html
- [Xu et al., ICML2021] http://proceedings.mlr.press/v139/xu21k.html
- [Pham & Nguyen, ICLR2021] https://openreview.net/forum?id=KvyxFqZS_D

### Clarity (Is the paper clearly written?)

- [Cl1] The paper is easy to read. I was able to understand the proof without much difficulty.
- [Cl2] I would suggest the overview of the proof strategy for the proof for deep linear GNNs in the main paper.

### Other comments

- l.73: shouldbe → should be
- l.108 : Eq. (5) I would like you to clarify with respect to which probability distribution the expectation $E$ in (5) takes. Is it the expectation value for the distribution of $x$ and not for the initial distribution of $W$?
- l.128: We, on the other hand are interested in ... → We, on the other hand, are
- l.176: $\sigma$ is used for both the variance of the Gassian distribution and the activation function.
- l.206: $L_j(w_{j, t}) \leq \varepsilon^2/h$ → $L_j(w_{j, T}) \leq \varepsilon^2/h$
- l.206: $|\beta_t| \leq \frac{\varepsilon}{4hn}$ → $|\beta_T| \leq \frac{\varepsilon}{4hn}$
- l.206: $\|\ell_t-1\| \leq \frac{\varepsilon}{4hn}$ → $\|\ell_T-1\| \leq \frac{\varepsilon}{4hn}$
- l.289, l.291: Same reference with different versions.

**Time Spent Reviewing:**

8

---

> ### Author Response · Authors · 2021-08-10
> **Response to reviewer iGLS**
>
> Thank you very much for your insightful review. We will take into account your suggestions and the additional references provided when preparing the final version. Below we address specific questions.
>
> **Q: [So1]:**
>
> **Response:** Thank you for asking the question. Yes, our analysis extends to the case when only a partial subset of the nodes are revealed. In eq. (5) we can define the squared loss over the subset of nodes revealed and the entire analysis works as is for this case as well. We are happy to clarify this further during the discussion stage.
>
> **Q: [So2]:**
>
> **Response:** Thank you for pointing this out! While the vanilla GCN architecture does not impose weight sharing, there are variants of graph neural networks used in practice that do enforce parameter sharing across layers especially for very deep networks (e.g. https://arxiv.org/abs/1511.05493). Our model is inspired by such variants (and the analysis is quite non-trivial even in this setting). We agree that this is only a subclass of GCNs. We will clarify this further in the paper. We believe however that our techniques can be extended to the case when the weights are not shared. This is a direction for future work.
>
> **Q: [So3]:**
>
> **Response:** Note that in the experiments we are simulating “exact” gradient descent. So the only source of randomness is the random initialization and there is no error in computing the gradients. Hence the plots closely resemble the expected convergence rates for the ReLU and deep linear networks. By re-doing the plots in log-scale we can see the difference where the convergence rate of the ReLU network is linear and the convergence rate of the deep linear network is exponential. We will add these plots in the final version as well.
>
> **Q: [C2]:**
> **Response:** Note that if the conditions on $\beta_t$ and $\ell_t$ hold for some $t$ then we already have a small loss value. Hence in the proof of Theorem 1 we show that for any $t$, either the condition on $\beta_t$ and $\ell_t$ hold (in which case the loss is at most $\epsilon^2$), or the PL condition holds in which case the loss decreases via the gradient update (eq (17)).
>
> **Q: [N4]:**
>
> **Response:** Thank you for pointing out the mean-field papers. Note that even in mean-field type analysis the width of the network tends to infinity. We, in particular, are interested in low width settings. For instance our learner network is of the same size as the teacher network (without any overparameterization). We will update the sentence accordingly.
>
> **Q: l.108 : Eq. (5):**
>
> **Response:** In eq. (5) $W$ is a fixed parameter vector and the loss of $W$ is calculated by taking the expectation over the distribution of $x$.

---

> > ### Comment · Reviewer_iGLS · 2021-08-27
> > **Thank you for your response.**
> >
> > I would like to thank the authors for taking my review comments seriously and responding to them. Let me confirm some parts of them that I do not understand.
> >
> > [So1] I agree with the authors in that the analyses of this paper can extend to the case where GNNs cannot observed all labels. I want to note that GNNs need to know features of ALL nodes. That is, we need to consider the transductive setting where a learner needs to know features of test instances.
> >
> > [So2] OK. I think the paper is significant in the sense of [N1]--[N4] even if the scope of this paper is restricted to the subset of GCN variants.
> >
> > [So3] Thank you for your comments. What I was wondering is not only whether the loss curve descrease polynomially or exponentially. Rather the rate is $O(1/\varepsilon^2)$ (ignoring log terms) in the one-layer ReLU-GNN setting. I agree with the authors that a log-scale plot can differentiate between exponential and polynomial decay and the rate of the decay. Can you see the rate is certainly $O(1/\varepsilon^2)$?
> >
> > [C2] OK.
> >
> > [N4] OK
> >
> > [l.108] : OK

---

> > > ### Author Response · Authors · 2021-08-29
> > > **Thanks you for your careful review**
> > >
> > > > [So1] I agree with the authors in that the analyses of this paper can extend to the case where GNNs cannot observed all labels. I want to note that GNNs need to know features of ALL nodes. That is, we need to consider the transductive setting where a learner needs to know features of test instances.
> > >
> > > We agree that our model does not apply to a semi-supervised/transductive setting where the graph of train+test nodes are known at train time, but neither the labels nor the features of the test nodes are known. But it works for semi-supervised settings where the features of the test nodes are known but not the labels. We will clarify this more in the final version.
> > >
> > > > [So3] Thank you for your comments. What I was wondering is not only whether the loss curve descrease polynomially or exponentially. Rather the rate is $O(1/\epsilon^2)$ (ignoring log terms) in the one-layer ReLU-GNN setting. I agree with the authors that a log-scale plot can differentiate between exponential and polynomial decay and the rate of the decay. Can you see the rate is certainly $O(1/\epsilon^2)$?
> > >
> > > From the loss curve on log scale we can differentiate between exponential and polynomial decay providing evidence that a convergence bound that depends on $\log(1/\epsilon)$ is not possible for the ReLU case. However we cannot definitively say that the polynomial decay is of the order of $1/\epsilon^2$.

---

### Official Review · Reviewer_3AyB · 2021-07-19

**Rating:** 6
**Confidence:** 4

**Summary:**

This is a theoretical paper analyzing the optimization of graph neural networks (GNN). It assumes that the data is generated from a GNN with unknown weights, and tries to recover it by the gradient descent on the population loss. The convergence rate is established for two settings: one-layer GNNs with ReLU activations, and deep linear GNNs.

**Limitations And Societal Impact:**

- As mentioned in the conclusions part, the extension of this approach to more general settings is still challenging.
- It would be better if Section 4 could show some proof sketch of the deep linear GNNs.

**Main Review:**

Originality: Previous optimization analysis usually consider fully connected networks or residual networks, and this paper considers GNNs that are proposed from applications with few theoretical results.

Quality: The theorems in the paper are supported by mathematical proofs. The authors consider some simplified settings: there exists a set of true weights; the gradient descent is taken on the population loss instead of empirical loss. Nevertheless, I think the proofs are not just trivial extension of fully connected networks, and it is a good starting point in this direction.

Clarity: The statements of the theorems and proofs are clear to me.

**Time Spent Reviewing:**

2

---

> ### Author Response · Authors · 2021-08-10
> **Response to reviewer 3AyB**
>
> Thank you for your encouraging review! We will take your suggestion on Section 4 into account and include a proof sketch for the deep linear case.

---

### Decision · Program_Chairs · 2021-09-27

**Decision:**

Accept (Poster)

**Comment:**

This paper analyzes convergence properties of gradient descent for graph neural networks by using the neural tangent kernel technique. More specifically, it shows convergence with iteration complexity $\epsilon^{-2}\log(1/\epsilon)$ for one hidden layer ReLU-GNNs and shows linear convergence for deep linear GNNs. The theory is verified through a numerical experiment.

In the previous researches, NTK is analyzed mainly for FNNs, but this paper extends it to GNN settings. This requires some technical novelty and is not trivial. The second analysis for the deep linear model is a bit restrictive because it requires linear activation and the parameters in all layers are the same. However, this also requires some technical novelty to extend the proof for FNNs to GNNs. In that sense, this paper has novelty.
The numerical experiments justify the theoretical results, but I recommend the authors that they present the graphs in log-log scale or semilog scale to see the correctness of the convergence rate given in the theorems.

Although there are some concerns as I stated above, this paper still possesses novelty and the convergence analysis for GNNs is an important issue in the literature. Hence, I recommend this paper for acceptance.